# End-to-End Premature Ventricular Contraction Detection Using Deep Neural Networks

**DOI:** 10.3390/s23208573

**Published:** 2023-10-19

**Authors:** Dimitri Kraft, Gerald Bieber, Peter Jokisch, Peter Rumm

**Affiliations:** 1Fraunhofer IGD Rostock, 18059 Rostock, Germany; dimitri.kraft@outlook.de (D.K.); gerald.bieber@igd-r.fraunhofer.de (G.B.); 2custo med GmbH, 85521 Ottobrunn, Germany; peter.jokisch@customed.de

**Keywords:** Ventricular premature contractions (PVC) detection, 1D U-Net neural network, Holter monitoring

## Abstract

In Holter monitoring, the precise detection of standard heartbeats and ventricular premature contractions (PVCs) is paramount for accurate cardiac rhythm assessment. This study introduces a novel application of the 1D U-Net neural network architecture with the aim of enhancing PVC detection in Holter recordings. Training data comprised the Icentia 11k and INCART DB datasets, as well as our custom dataset. The model’s efficacy was subsequently validated against traditional Holter analysis methodologies across multiple databases, including AHA DB, MIT 11 DB, and NST, as well as another custom dataset that was specifically compiled by the authors encompassing challenging real-world examples. The results underscore the 1D U-Net model’s prowess in QRS complex detection, achieving near-perfect balanced accuracy scores across all databases. PVC detection exhibited variability, with balanced accuracy scores ranging from 0.909 to 0.986. Despite some databases, like the AHA DB, showcasing lower sensitivity metrics, their robust, balanced accuracy accentuates the model’s equitable performance in discerning both false positives and false negatives. In conclusion, while the 1D U-Net architecture is a formidable tool for QRS detection, there’s a clear avenue for further refinement in its PVC detection capability, given the inherent complexities and noise challenges in real-world PVC occurrences.

## 1. Introduction

Premature ventricular contractions (PVCs) are a prevalent cardiac arrhythmia frequently associated with an array of severe cardiovascular diseases. The swift, precise, and real-time identification of these events is pivotal for timely clinical interventions. With the increasing ubiquity of consumer-grade cardiac monitoring devices, there is an urgent demand for sophisticated detection algorithms that can operate effectively on these devices. Importantly, such an algorithm needs to handle the limitations of single-channel data, effectively mitigate noise interference, and function independently of cloud resources.

In this publication, we introduce a novel approach to PVC detection, conceptualizing the task as a one-dimensional (1D) segmentation problem. We employ a deep learning model rooted in the U-Net architecture [1]. Originally conceived for biomedical image segmentation, the U-Net model has demonstrated its adaptability and efficiency across a range of applications. In our research, we repurpose it to process 1D electrocardiogram (ECG) signals, underscoring its potential for robust, end-to-end PVC detection. Our objective was to devise an algorithm with the capability to consistently recognize the intra- and interindividual variability that PVCs can exhibit.

This study’s emphasis is on the model’s ability to process noisy, single-channel ECG data commonly obtained from consumer-grade devices. We evaluate our single-channel model on two- and three-channel ECG data using a simple merging technique of beat lists. Processing a single ECG at a time may have multiple advantages over processing multiple leads at once. We see the following advantages:Simplicity: Single-channel ECG processing is simpler and more straightforward, which can make the development and debugging of algorithms easier.Data Availability: In some situations, only single-lead ECG data might be available (e.g., Icentia 11k DB). Many portable and wearable ECG devices only record a single lead, so algorithms designed for single-lead data can be more broadly applied.Robustness to Noise: Single-lead ECGs might be less susceptible to noise and artifacts that can affect multilead recordings. For instance, movement artifacts can affect different leads to different extents, potentially making multilead data more challenging to interpret. By analyzing each lead independently, we may overcome this.

In the context of single-channel ECG data, which often harbor a significant amount of noise, our model stands out by precisely detecting both normal and PVC beats—a marked enhancement over prevailing methodologies. Notably, our approach is tailored to function exclusively on physicians’ existing hardware, obviating the need for cloud support. This strategic decision not only circumvents latency issues associated with cloud-based processing but also reinforces data privacy, enabling swift ECG analysis.

The core contributions of our study are:The development of an intuitive framework for end-to-end beat classification, sidestepping the conventional need to segment each beat individually.The deployment and comprehensive evaluation of a machine learning algorithm across a variety of datasets, each with its unique set of challenges.The achievement of superior performance metrics across these datasets, pushing beyond established state-of-the-art benchmarks.The crafting of an efficient model optimized for offline processing, mitigating the need for hefty computational resources.

## 2. Related Work

The application of neural networks for automated detection of cardiac arrhythmia such as PVCs has been the subject of research for a number of years, with various models demonstrating improved outcomes over conventional signal processing techniques. These studies provide valuable insights for our current research.

One of the earlier significant works was presented by Kiranyaz, Ince, and Gabbouj [2], who developed a one-dimensional convolutional neural network (1-D CNN) for patient-specific ECG classification, inclusive of PVC detection. Despite its impressive adaptability to unique ECG patterns of individual patients, the need for patient-specific training may hinder its scalability and applicability to real-time detection, especially on consumer-grade devices.

Following this, Acharya et al. [3] proposed a deep convolutional neural network (CNN) for automated detection of cardiac disorders, including PVCs. The model was trained using single-lead ECG signals, showing promising outcomes, but handled only evaluated high-frequency noise (Gaussian).

More recently, Hannun et al. [4] developed a deep learning model referred to as the Cardiologist-Level Arrhythmia Detector (CLAD), which employed a 34-layer CNN for multiclass detection of 14 types of cardiac arrhythmia from single-lead ECG records, including PVCs. While CLAD demonstrated remarkable performance, its high complexity could potentially limit its implementation on consumer-grade devices with constrained computational resources. Moreover, it was not specifically tailored for PVC detection.

While 1D ECG arrhythmia detection approaches achieve good performance, 2D image-based ECG detection methods also exist [5]. Either an image of a ECG recording or the analysis of a 2D spectrogram [6] is widely used to approach the detection of arrhythmia.

The authors of [7] presented an approach for PVC classification that leverages a straightforward support vector machine (SVM), achieving commendable sensitivity and specificity rates around 99%. Their methodology encompasses a range of preprocessing and feature extraction procedures. Nevertheless, there is a noteworthy detail in their experimental design that may influence the interpretation of these results. The team employed 10% of the PVC beats from the MIT dataset for training purposes, the very dataset they subsequently used for validation. Given the nature of the MIT dataset, which is characterized by substantial intersubject variability contrasted by minimal intrasubject variability, this could potentially introduce the risk of overfitting and biased results. The issue is further compounded in real-world applications, where ECG patterns showcase higher complexity and diversity. This raises concerns about the model’s robustness and its ability to generalize in such scenarios. Interestingly, these reservations seem to gain some traction when we consider the performance drop 2% seen on their experimental dataset, which comprises only 903 PVC beats.

We discuss multiple additional issues related to PVC detection in the current literature:Real-world settings are not considered: Electrocardiogram (ECG) data inherently contain various types of noise, including baseline wandering, power line interference, muscle noise, and other artifacts related to contact with the electrodes. These noise elements pose significant challenges to the extraction of robust features, consequently affecting the performance of PVC classification in real-world settings. Thus, an algorithm that performs well on a clean, noise-free dataset may not perform as well when deployed in a real-world setting where the noise level is higher or varies unpredictably.Testing datasets are not representative: Gender differences in ECG are well documented in the literature. Men and women can have different heart rates, QRS complex durations, QT intervals, and T-wave morphologies, among other characteristics. These differences can affect the performance of PVC detection algorithms if they are not properly accounted for during algorithm development and testing [8,9].Training and testing datasets are not separated: A notable limitation of many existing methods lies in their reliance on small or overlapped ECG datasets for training and testing. This practice raises questions about their efficiency and generalizability when applied to a large collection of ECG recordings—an issue that remains largely unaddressed in the literature [10].

In the realm of black-box AI methodologies, a significant concern is the inherent lack of interpretability, which often acts as a barrier to the adoption of AI-driven solutions in clinical settings. Various initiatives have sought to demystify the internal workings of these AI models. For instance, Bender at al. [11] examined the decision-making process of the network by analyzing input gradients. An alternative strategy emphasizes the creation of transparent network constructs—be they architectures, blocks, or layers—wherein both activations and trainable parameters possess tangible, physical interpretations [12].

In light of these existing studies, our research aims to bridge the gap by delivering a robust, efficient, and highly accurate model for PVC detection that is specifically tailored for noisy, single-channel ECG data from consumer-grade devices. The novelty of our approach lies in its unique application of the U-Net architecture to the problem of PVC detection as a 1D segmentation task. Moreover, our model operates independently of cloud resources, ensuring real-time detection with improved data privacy.

### 2.1. Beat Detection Performances

The accurate detection of normal cardiac beats is essential for the diagnosis and monitoring of cardiovascular diseases. Various algorithms have been proposed over the years, with each aiming to optimize the detection accuracy under different conditions. The Table 1 presents a comparative assessment of several popular normal beat detection algorithms applied to the MIT-BIH dataset, a well-regarded benchmark for cardiac signal analysis. Both the complete MIT-BIH dataset and a version with VFib beats excluded are considered. Performance metrics including sensitivity (Se), positive predictive value (PPV), and F1 score are used to gauge each algorithm’s efficacy. This table was adapted from [13], providing a consolidated view of the advancements in this domain.

### 2.2. PVC Detection Performances

Premature ventricular contractions (PVCs) are early heartbeats originating in the ventricles of the heart. Their accurate detection is critical, given their potential association with various cardiac disorders. The Table 2 offers a comparative study of several prominent PVC beat detection algorithms when applied to a subset of 11 records from the MIT-BIH dataset, a renowned benchmark in cardiac signal processing. This subset provides a specific environment to evaluate the algorithms’ performances due to the unique characteristics and challenges posed by PVC beats. The included performance metrics are sensitivity (Se), positive predictive value (PPV), and F1 score. This table adapted from [13] enables readers to comprehend the current state of the art in PVC beat detection and the relative efficacy of different methods.

## 3. Methods

In the methodology section of this paper, we delve into the specific techniques utilized for the detection and differentiation of normal heartbeats and premature ventricular contractions (PVCs). Given the critical implications these heart rhythms have in clinical practice and patient care, the accuracy and efficiency of these detection algorithms are paramount. Our methodological approach incorporates a broad range of computational tools and techniques, each tailored to handle the specific characteristics of normal heartbeats and PVCs. These include but are not limited to data preparation, signal preprocessing, machine learning algorithms, and evaluation metrics.

### 3.1. Overview

In our study, we employed a 1D U-Net architecture [1] for end-to-end detection of premature ventricular contractions (PVCs). The U-Net consisted of four encoder layers, one bottleneck layer, and four decoder layers. The number of filters in the initial layer was set to 16 and was multiplied by 2 after each encoder layer. The U-Net convolution blocks used had a filter size of 9 and a dilation of 3. Instead of the conventional max-pooling operations in the encoding stages, we used strided convolutions, and each convolution was followed by a rectified linear unit (ReLU) activation function.

For the training of the U-Net, a composite loss function comprising dice loss [28] and focal loss [29] was used, with equal weights of 0.5 assigned to each loss. The focal loss incorporated alpha coefficients of 1 (background), 1 (normal), and 1.5 (PVC). An initial learning rate of 0.001 was set, and the network was optimized using the Adam optimizer [30] with beta values of 0.9 and 0.999 and no weight decay.

Our training dataset comprised the St. Petersburg 12-channel ECG dataset [31] (https://physionet.org/content/incartdb/, (accessed on 15 August 2023)), the Icentia 11k dataset [32] (https://physionet.org/content/icentia11k-continuous-ecg/, (accessed on 15 August 2023)), and a custom dataset that was meticulously curated by the authors, containing approximately 7500 single-channel ECG windows ranging from 10 to 30 s each. This custom dataset, crafted to encapsulate challenging real-world scenarios, emerged from comprehensive data gathering and curation efforts and is not available to the public. It offers a unique set of examples distinct from what is typically found in standard public datasets. We targeted the detection of normal and PVC heartbeats. We designed a segmentation mask where class 1 was assigned to normal beats within a window of 200 ms around the R peak, while class 2 was assigned to PVCs detected 100 ms before and 150 ms after the R peak. The background was labeled as class 0. Figure 1 shows the general approach for the detection of PVC and normal beats using a 1D U-Net.

### 3.2. Data Preparation

The ECG signals were divided into discrete 30 s segments, without any overlap, and a corresponding segmentation mask was developed for each of these sections as previously described (see Figure 2). If the ECG signals were less than 30 s in length, they were zero-padded to reach the desired 30 s duration. We made the decision not to normalize the amplitude of the ECG signals, opting instead to apply certain preprocessing steps. These included the utilization of a high-pass forward–backward Butterworth filter with a set cutoff frequency of 0.5 Hz, as well as a power line filter. To further refine the signals, all were downsampled to a frequency of 125 Hz using linear interpolation. These preprocessing steps were executed on the entirety of the signal prior to division into the aforementioned 30 s segments. This procedural routine was consistently applied across all the datasets. The samples we selected only contained either a normal or a premature ventricular contraction (PVC) beat. We consciously omitted windows that contained beats that could not be accurately classified during dataset labeling. Figure 3 displays a representative 10 s ECG sample paired with its corresponding segmentation mask for illustrative purposes. The construction of the segmentation mask is delineated below. For each regular beat across our datasets, a label of class index 1 is assigned 100 ms preceding and following the R-peak. For PVCs, a label of class index 2 is given 100 ms before the R peak and extended to 150 ms post R peak to encompass the entire characteristic of a PVC beat. Any other region is labeled with a 0, denoting the background (see Figure 3b). The compilation of the training set resulted in approximately 4 million 30 s windows. To tackle the prevalent class imbalance often seen in medical datasets, we opted for an oversampling approach tailored to the PVC class during training. While the focal loss function was used to mitigate class imbalance effects, it doesn’t fully prevent potential biases towards the dominant class. Therefore, to foster unbiased learning and bolster our model’s PVC detection efficiency, we aimed to achieve an equitable distribution of both PVC and regular heartbeat classes in our training set.

### 3.3. Model Architecture

In our methodology, we employed a conventional 1D U-Net architecture for ECG consisting of 4 encoder layers, a single bottleneck layer, and 4 decoder layers. We chose a one-dimensional architecture to process ECG data. This allows for analysis of single- and multichannel ECG data and is not limited to a particular hardware constellation. Although simultaneous inclusion of all channels would be desirable, a 1D architecture makes preprocessing and explainable AI more appropriate. In addition, a multichannel ECG may provide low-noise signals on one channel while the others are of poor quality. This interferes with simultaneous analysis of the total dataset. Therefore, a 3D architecture would be more error-prone or complex to design, while a 1D architecture is fast and efficient and can easily handle and identify noisy or corrupt data. We consciously decided not to incorporate self-attention layers, preferring to retain standard dilated convolution layers. The rationale behind this decision was to enable efficient real-time inference on consumer-grade devices, which is crucial for practical applicability. Moreover, we intentionally developed a localized solution, given that the transmission of sensitive health data to a cloud or server is generally not endorsed by the majority of cardiologists in Germany due to privacy concerns. U-Net is composed of two parts: the “contracting path” (or encoder) and the “expanding path” (or decoder). The bottleneck layer is the layer that connects these two paths. In a traditional U-Net architecture, the bottleneck consists of two convolutional layers followed by a ReLU activation function. This layer serves as a bridge between the encoder and the decoder, reducing the spatial dimensions of the input data and allowing the network to focus on the most important features. It provides an abstracted, high-level understanding of the input. The encoder layers capture the context in the ECG, while the decoder layers enable precise localization using upsample layers and convolutions. We employ upsampling layers followed by a convolutional layer for two primary reasons. Our experiments indicated that utilizing a transposed convolutional layer both reduced our method’s accuracy and increased the parameter count. Our convolutional blocks consist of standard convolutional layers with a kernel size of 9, a dilation of 3, and the same padding. The second convolution layer in a block also has a stride of 2. We further use 1D batch normalization [33] and a ReLU activation function. Owing to the effect of dilation, we effectively utilize a receptive field of 25, which results in approx. 200 ms in the first layer. The output layer contains a 1 × 1 convolution that maps the 16 output channels from the previous convolutional layer to an L × C output matrix, where L is the input length of the ECG and C is the number of classes. All hyperparameters were identified using an extensive grid search. The architecture is depicted in Figure 4.

### 3.4. Augmentation

During the training phase, a strategic augmentation of data was undertaken to promote a higher degree of model generalizability and resilience against variations in real-world scenarios. This augmentation process incorporated the random scaling of signal amplitudes; the infusion of random Gaussian, pink, and brown noise; and the induction of minor baseline shifts. Notably, we refrained from employing other prevalent augmentation procedures such as signal masking, temporal shifting, time compression and stretching, mixup [34], and cutmix [35].

To evaluate the effect of data augmentation, we applied a combination of the following transformations dynamically during training:Scaling of the amplitude with a probability of p=0.75 and a scaling factor of [0.6,…,1.4];Offset of the amplitude with a probability of p=0.75 and an offset value of [−0.2,…,0.2];Addition of Gaussian, brown or pink noise with a probability of p=0.75 and an offset value of [−0.2,…,0.2].

Although data augmentation does not invariably lead to enhanced performance in practical scenarios, certain methods can adversely affect outcomes, as highlighted by Raghu et al. [36] in the context of AFib detection. However, based on our trials, the trio of signal transformations discussed previously proved optimal for our distinct task and dataset. This aligns with Rahman et al.’s systematic review [37] on ECG signal data augmentation.

### 3.5. Post Processing

The model outputs are processed using the softmax function, an exponential function that generates three float values in the range of 0.0 to 1.0, summing to 1.0. These values can be interpreted as the probability that the corresponding time point lies within no beat, a normal beat, or a PVC (premature ventricular contraction). When the first, second, or third values of the triples are grouped together as a vector, three masks are created:The no-beat mask: the probability that each time point does not lie within any beat;The normal-beat mask: The probability that each time point lies within a normal beat;The PVC Mask: The probability that each time point lies within a PVC.

The PVC (premature ventricular contraction) list is created by scanning the PVC mask until a threshold of 0.5 is reached or exceeded, marking the potential onset of a PVC. The termination of the potential PVC is detected when the threshold of 0.5 is consistently undershot for at least 2 samples. A PVC is acknowledged if the period between the onset and termination of the threshold exceedance lasts for a minimum of 5 samples (equivalent to 40 ms). The midpoint between the onset and termination of the exceedance region is selected as the point of the beat. The list for normal beats is created in an analogous manner. Following this, the two lists are merged. Lastly, beats that lie within regions that clearly do not contain ECG data, such as at the end of the evaluation when electrodes have already been removed and where the neural network might have erroneously detected pseudobeats, are canceled.

### 3.6. Training Data

We utilized three different datasets as training data. The first one is a custom 3-channel ECG dataset collected from a wide variety of subjects. The second dataset is the Icentia 11k dataset, consisting of single-channel ECG data from 11,000 subjects, representing a large dataset for the training of an end-to-end PVC detection neural network. The third dataset is the 12-channel St. Peterburgs INCART dataset.

#### 3.6.1. Custo Med Training Dataset

We utilized a custo med flash 500/510 3-Channel Holter (see Figure 5), with a sampling frequency of 125 Hz and a 5.6 microvolt/Bit resolution with 10 bit resolution.

These data were obtained in an anonymized form from one of our clients. As such, we do not possess information regarding the age and sex of the individuals associated with the electrocardiograms. For the training data, approximately 1000 ECGs were employed, albeit not all from unique patients. From these ECGs, we generated between 3 and approximately 30 snippets of varying lengths (ranging between 10 s and around 120 s), which resulted in 7500 ECG samples. These ECG snippets underwent careful evaluation, with corrections made to annotations as necessary to ensure accuracy. In addition, we utilized 36 different 24 h ECG recordings taken from 36 unique patients for an extended, long-term monitoring evaluation. This dataset was designed to focus on QRS and PVC classes specifically; thus, only these were annotated (see Table 3).

#### 3.6.2. Icentia 11k

The Icentia 11k dataset was primarily designed for unsupervised representation learning for arrhythmia subtype discovery. The data originate from CardioSTAT, a single-lead heart monitoring device developed by Icentia [32]. This device records raw signals at a 250 Hz sampling rate with a 16 bit resolution in a modified lead 1 position. The dataset is compiled from the records of 11,000 patients predominantly from Ontario, Canada, who were monitored using the CardioSTAT device across various medical centers. The data underwent analysis by a team of 20 technologists from Icentia using proprietary analysis tools. Initial beat detection was conducted automatically, following which a technologist manually examined the record, labeling beat types and rhythms by conducting a full disclosure analysis, wherein the entire recording is assessed. Each analysis underwent final approval by a senior technologist prior to inclusion in the dataset. The average age of the patients was 62.2 ± 17.4 years. The Icentia 11k dataset consists of 2 billion normal and 17 Million PVC beats (see Table 4). Therefore, we subsampled the dataset, considering only 1% of segments with normal beats and 100% of the segments in which at least one PVC beat was present. Since PVC beats are rare in the dataset (less than 1%), we mostly settled on segments containing normal beats with little variation.

#### 3.6.3. St. Petersburg INCART 12-Lead Arrhythmia Database

This database is composed of 75 annotated recordings derived from 32 Holter records. Each recording extends for a duration of 30 min and comprises 12 standard leads. These leads were sampled at a rate of 257 Hz, with the gains fluctuating between 250 and 1100 analog-to-digital converter units per millivolt. The database’s reference annotation files collectively contain more than 175,000 beat annotations.

The original records were gathered from patients undergoing tests for coronary artery disease. In terms of demographics, 17 patients we male, and 15 were female, ranging between the ages of 18 and 80, with a mean age of 58. All patients in the cohort were devoid of pacemakers; however, most presented with ventricular ectopic beats. The records selected for inclusion in the database were primarily those of patients demonstrating ECG patterns indicative of ischemia, coronary artery disease, conduction abnormalities, and arrhythmia. Key observations from the selected records are reported as follows (Table 5):

Annotation of the data was initially performed by an automated algorithm, which was then manually rectified in accordance with standard PhysioBank beat annotation definitions. The algorithm typically positions beat annotations at the center of the QRS complex, as deduced from all 12 leads. Despite this, these locations did not undergo manual corrections, which might lead to occasional misalignments in the annotations.

### 3.7. Test Data

The test data consist of publicly available and private datasets. No testing data are included in the training data, so we have a fairly good understanding of the generalizability of our model. We utilized the standard MIT (https://www.physionet.org/content/mitdb/1.0.0/, (accessed on 15 August 2023)) and MIT 11 subset datasets, the AHA dataset https://www.ecri.org/american-heart-association-ecg-database-usb, (accessed on 15 August 2023)), and the NST dataset https://physionet.org/content/nstdb/ (accessed on 15 August 2023)) to evaluate the performance of the model under noisy conditions and our own collected datasets, i.e., CST and CST Strips, where CST DB is a dataset consisting of 18 records of 24 h ECG samples and CST Strips contains 628 difficult real-world ECG data samples (noisy, containing couplets and triplet PVCs and other anomalies) from different subjects.

#### 3.7.1. AHA

The American Heart Association database (AHA database) is not available for download from PhysioNet or similar platforms but can be exclusively obtained from ECRI. Established during the late 1970s and early 1980s through a collaboration between the American Heart Association and the National Heart, Lung, and Blood Institute, the AHA Database serves as a valuable resource for the evaluation of ventricular arrhythmia detectors. Completed in 1985 without subsequent updates, it comprises 80 two-channel analog ambulatory ECG recordings. These were digitized at a 250 Hz per channel frequency and a 12 bit resolution over a 10 mV range and classified into eight categories based on the severity of ventricular ectopy. These categories range from ‘no ventricular ectopy’ to ‘ventricular flutter/fibrillation’. Each recording contains a thirty-minute segment annotated beat by beat, without differentiating supraventricular ectopic beats from normal sinus beats. Two versions of the database (short and long) are available, with the latter including 2.5 h of unannotated ECG signals preceding each annotated segment.

In addition to the initial dataset, there exists a second set of 75 recordings (test set) assembled using identical criteria. Reserved for evaluations without the detectors being tuned for the test data, this test set was made available around 2003. The nomenclature of the records in the test set parallels that of the development set, with the first digit denoting the class and the second indicating the version (1 for the long version and 3 for the short) [31]. The AHA datasets contain a wide spectrum of beats, but we focus on the detection of normal and PVC beats. Therefore, our AHA test dataset contains only N and V beats (see Table 6).

#### 3.7.2. NST

The dataset includes 15 recordings, each with a half-hour duration, comprising 12 electrocardiogram (ECG) recordings and 3 recordings characteristic of noise typically found in ambulatory ECG recordings. The noise recordings comprised baseline wander, muscle artifacts, and electrode motion artifacts, each obtained from physically active volunteers using a standard ECG apparatus. The ECG recordings were generated from two clean ECGs (118 and 119) from the MIT-BIH Arrhythmia Database. Noise from the electrode motion artifact record was then artificially incorporated into these clean ECGs to simulate real-world noisy conditions. This noise was added after the first 5 min of each ECG, alternating between two-minute segments of noise and clean intervals. The records have varied signal-to-noise ratios (SNRs) ranging from 24 dB to −6 dB during the noisy segments to test algorithms under different noise conditions. As the original ECGs are noise-free, the accurate beat annotations are known, and these annotations serve as the reference, even in instances in which the noise renders the ECGs visually unreadable [38]. Note that we utilized all noise levels in the dataset, unlike other works, e.g., [39].

#### 3.7.3. MIT

The MIT-BIH Arrhythmia Database comprises 48 two-channel ambulatory ECG recordings, with a half-hour duration, from 47 subjects evaluated by the BIH Arrhythmia Laboratory between 1975 and 1979. These recordings were selected both randomly and purposefully from a larger set of 4000 records of 24 h ECG recordings collected from a diverse patient population at Beth Israel Hospital. The digitization of these recordings was performed at a rate of 360 samples per second per channel with an 11 bit resolution over a 10 mV range. Each record was annotated independently by two or more cardiologists to create computer-readable reference annotations for each beat. As of February 2005, all 48 complete records and reference annotation files were freely available.

The MIT testing dataset consists of 24.07 h of data. We further utilized a subset of the MIT dataset, the MIT 11-record subset, which is also widely used to evaluate performance. This dataset consists of around 5.52 h of ECG data from 11 different patients. For our evaluation, we only utilized 44 samples, since 4 of them were not adequate for processing. The MIT Arrhythmia test dataset contains a wide spectrum of beats, but we focused on the detection of normal and PVC beats. Therefore, our MIT test dataset contains only N and V beats (see Table 7) [40].

#### 3.7.4. Custo Med Test Dataset

The CST Strips dataset consists of 627 records of 10 to 30 s ECG segments. The data were sampled from multiple patients, with a strong focus on noisy, difficult-to-classify PVC beats, couplets, triplets, or salves (4756 PVC beats). The number of normal and PVC beats may be found in Table 8.

## 4. Results

The quantitative results on our test data are outlined in detail in Table 9. We utilized sensitivity (se), specificity (sp), and balanced accuracy (ba) to assess the performance of our model. We surpassed the state of the art on the MIT 11 dataset and achieved stable performance across all datasets.

Sensitivity, also known as the true-positive rate, is calculated as the proportion of actual QRS complexes or PVCs that the algorithm correctly identifies as such. A high sensitivity means the algorithm is good at catching these events, reducing the number of false negatives (instances where a QRS complex or PVC is present but the algorithm fails to detect it). We calculated the TN based on whether a certain beat is outside the range of 150 ms, matching a corresponding RR peak.

Our dataset is strongly imbalanced towards the normal beat class (QRS). In the context of classification tasks, imbalanced datasets are those in which the classes are not represented equally. For instance, in a binary classification task, you might have a dataset with 95% examples of class A and only 5% examples of class B. In such cases, a model might achieve high accuracy by simply predicting the majority class. For example, a model that always predicts class A would be 95% accurate, which could lead to a misleading picture of its effectiveness. In these scenarios, balanced accuracy comes to the rescue. It is the average of sensitivity (the true-positive rate or the proportion of actual positives correctly identified) and specificity (the true-negative rate or the proportion of actual negatives correctly identified), so it takes into account both false positives and false negatives. By doing this, it provides a more fair view of the model’s performance across all classes rather than favoring the majority class. The metrics are calculated as follows:(1)$$Sp=\frac{TN}{TN+FP};\;Se=\frac{TP}{TP+FN};\;BA=\frac{Se+Sp}{2}$$

### 4.1. Evaluation Method

The objective of our evaluation is to generate a list of PVC beat locations for each recording that align with the ground truth PVC annotations. For every reference PVC annotation, there should be a corresponding predicted PVC annotation within a 150 ms interval centered around it. Note that reference PVC annotations present in the first or last 0.2 s of the recording are disregarded. Any detected PVC should fall within 150 ms of its reference annotation. This procedure is also applied to the normal beat locations.

### 4.2. Model Output

The output of the system is a series of classifications for each time stamp in the input ECG data as either a background, normal, or PVC beat (see Figure 6). This end-to-end process allows for the automatic detection of PVCs from ECG data, reducing the need for expert manual review and potentially speeding up the diagnostic process. For a given variable-length ECG, the model outputs a corresponding equal-length segmentation mask. The minimum length of the ECG signal is around 400 input time stamps, which, in our case (signal sampled at 125 Hz), equals 400/125 = 3.2 s of input data due to the kernel size of 9, the convolutional stride of 2, the same padding, and a dilation of 3.

### 4.3. Productive Usage

We incorporated our model into the custo diagnostic v5.9, MedTech+Science GmbH, Ottobrunn, Germany (see Figure 7) using the ONNX Runtime library for C++. The model size is around 13 MB, consisting of 836,387 trainable parameters and requiring 63 GFLOPs (estimated using fvcore (https://github.com/facebookresearch/fvcore), (accessed on 18 August 2023)) for 1 h of one-channel ECG data. A somewhat new CPU may run 60 or more GFLOPs, which results in around 48 s for a 24 h ECG. We tested the model on a i7-10750H, where the model requires 60 s for a three-channel 24 h ECG. For comparison, a standard 2D U-Net for aerial image segmentation may require 16 GFLOPs for a single 256 × 256 pixel grayscale image [41].

## 5. Model Interpretability

We utilized layer-wise gradient-weighted class activation mapping (LayerGradCAM) [42] as a visualization technique to highlight the important features in an input that contribute to the network’s final decision. In the context of ECG data, LayerGradCAM can provide significant insights into which ECG segments are most influential for the model’s segmentation decisions. In the ECG domain, it is often crucial to understand not just the final output of a model but also the ‘why’ behind that output. For instance, in the case of electrocardiogram (ECG) data, understanding which segments of the time series (i.e., specific heartbeats or signal patterns) are indicative of a certain disease can help clinicians make better-informed decisions. LayerGradCAM provides such an understanding by producing a heat map of the input time series, with the intensity of the color (in our case, green) indicating the contribution of each time point to the final decision (see Figure 8. This approach helps to uncover the model’s internal decision-making process, thereby increasing the interpretability and transparency of the model. Our primary insights obtained using LayerGradCam can be summarized in three main points:The R peak emerges as a significant determinant for the neural network’s assessments.When evaluating PVC beats, the neural network extends its focus beyond just the attributes of the PVC beat, also accounting for the features of adjacent beats.In the presence of noise, disturbances that mimic a QRS complex have the potential to be erroneously detected as either a normal or PVC beat.

We recognize the existence of various other methods for interpreting the decision-making process of neural networks, such as integrated gradients [43] and DeepLift [44]. Each of these approaches offers unique insights into the workings of a neural network.

## 6. Discussion

In the analysis of the different databases, several key observations emerge. The QRS detection specificity and sensitivity results are consistently high across all databases, with the specificity exceeding 0.95 and sensitivity exceeding 0.89 in all cases. This indicates a strong performance in the detection of QRS complexes. Comparatively, the detection of PVCs exhibits some variability. While the specificity remains high, the sensitivity fluctuates between 0.857 (AHA DB) and 0.991 (MIT 11 DB). Such a disparity may reflect the inherent complexity and variability of PVC occurrences in ECG recordings, requiring more nuanced detection algorithms. Note that the AHA database has several false annotations and also contains pacemaker and AFIB sections. The balanced accuracy (BA) for QRS detection is exemplary across all databases and closest to the ideal score of 1. This reflects an overall well-balanced performance between sensitivity and specificity, suggesting that the model does not lean excessively towards recall or precision. The lowest BA for QRS detection is seen in the NST DB (0.924), and the highest BA is observed in the MIT 11 DB (0.999). PVC detection, on the other hand, showcases a broader range in terms of balanced accuracy, from 0.909 (NST DB) to 0.986 (MIT 11 DB). It is noteworthy that despite the comparatively lower sensitivity in the AHA DB for PVC detection, its balanced accuracy still remains robust (0.915), demonstrating the model’s relative equilibrium in handling false positives and negatives. In our study, we observed an improvement over the state of the art for PVC beat detection using both the MIT DB and the MIT 11 DB. Specifically, we achieved a score of 0.986, slightly edging out the 0.984 reported in [13]. Regarding normal beat detection, our result of 0.998 for the MIT DB is on par with the findings presented in [18]. However, it is crucial to emphasize that these comparisons are pertinent only to the MIT DB.

Overall, these results highlight the robustness of QRS detection across multiple databases and demonstrate a degree of variability in PVC detection, underscoring the need for additional algorithm refinement for this latter category. The lower performance metrics in the NST DB and AHA DB relative to the other databases might indicate that the model does not perform optimally under strong noise conditions, suggesting an avenue for future investigation and algorithm optimization.

## 7. Limitations

While our study provides novel insights and adds value to the current literature, it is not without its limitations, which offer avenues for future work.

Pruning: Our model did not incorporate pruning techniques during the training process. Pruning is a common strategy to reduce the complexity and size of deep learning models, improving computational efficiency and potentially reducing overfitting. Future studies might explore the impacts of various pruning techniques on model performance and efficiency.Absence of an attention mechanism: The model did not leverage any attention mechanism. Attention models have emerged as powerful tools in deep learning, enabling the model to focus on the most relevant parts of the input for a given task. Incorporating attention mechanisms could improve the model’s performance, especially in tasks where certain parts of the input carry more informative content.Lack of self-supervised pretraining: Our study did not exploit self-supervised pretraining using multiple datasets. This approach could potentially improve the robustness and generalizability of the model by exposing it to a wider range of data during pretraining.Limited classification: The scope of our model was confined to the detection of normal beats and premature ventricular contractions (PVCs). Although this focus has its own merits, the model’s utility could be enhanced by expanding its classification capabilities to detect other types of cardiac events.Size of the test datasets: Our test datasets were not particularly large. Larger test datasets would provide a more robust estimation of the model’s performance and its ability to generalize to unseen data.Single-channel model: Our model was designed to work with single-channel ECG signals. While this design decision simplifies the model and its input requirements, it might limit the model’s ability to detect cardiac events that are better-characterized using multichannel ECG signals. Future research could investigate the benefits of a multichannel approach.

## 8. Conclusions

This study presents a comprehensive analysis of ECG signal processing for the detection of normal heartbeats and premature ventricular contractions (PVCs). The employed methodologies combine state-of-the-art machine learning approaches, particularly convolutional neural networks, which showed promising results in extracting intricate patterns from the ECG signals, despite the inherent challenges of noise and non-stationarity. Several databases were employed to test and validate the proposed methods, including the MIT DB, MIT 11 DB, AHA DB, NST DB, CST DB, and CSTStrips DB. Each database presented unique challenges and nuances that offered valuable insights into the robustness of our methodologies. The results were presented with a focus on sensitivity (Se), specificity (Sp), and balanced accuracy (BA)—metrics that are critical for evaluating the performance of any diagnostic tool. In our analysis across various databases, QRS detection showcased consistently high specificity and sensitivity, with values surpassing 0.95 and 0.89, respectively. Conversely, PVC detection displayed variability, especially in sensitivity, which ranged from 0.857 (AHA DB) to 0.991 (MIT 11 DB). The disparities might be attributed to the intricate nature of PVCs and potential inaccuracies within the AHA Database. Balanced accuracy for QRS detection remained impressive across all databases, approaching the ideal score, whereas PVC detection showed a wider range. Notably, despite the lower sensitivity of AHA DB for PVCs, its balanced accuracy was commendable, at 0.915. In summary, our results underline the reliability of our QRS detection and suggest the need for further refinement for PVC detection. The model’s challenges with NST DB and AHA DB might hint at its sensitivity to significant noise, pointing to potential areas for future enhancements. We further analyzed ECG recordings from 36 different patients for a long-term monitoring evaluation. This unique dataset allowed us to simulate real-world scenarios and to evaluate the potential of our methodologies in clinical settings. For the successful deployment of the model in a production environment, special attention was paid to developing an end-to-end framework, which streamlines the process from raw ECG input to PVC detection. This framework would serve as a foundation for the seamless integration of the model into existing health monitoring systems. While the obtained results are encouraging, the complexity of ECG signal processing and the inherent noise and variability in ECG data always leave room for future improvements. As such, this research is part of an ongoing process of refining and expanding our methods to develop the most effective tools for PVC detection and, more broadly, for cardiovascular health monitoring. It is our hope that this research will contribute improvements in the accuracy and reliability of ECG-based PVC detection, ultimately supporting the development of more effective personalized treatments for patients with cardiovascular conditions.

## Figures and Tables

**Figure 1 sensors-23-08573-f001:**
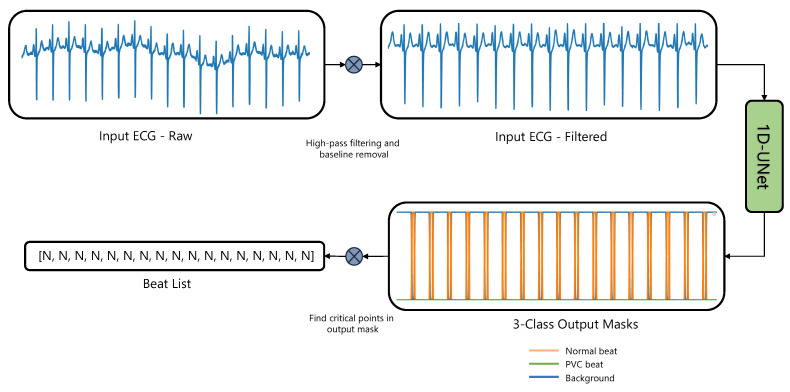
Diagram illustrating our method for end-to-end premature ventricular contraction detection with single-channel ECGs. For multilead ECG analyses, each channel is processed independently, with beat lists from all channels subsequently integrated via a majority voting mechanism.

**Figure 2 sensors-23-08573-f002:**
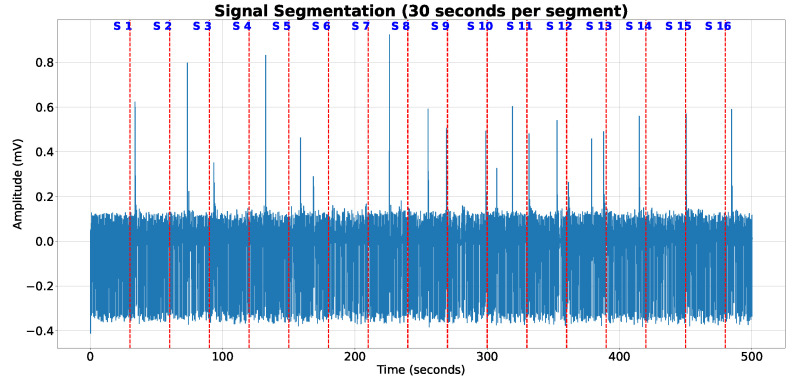
Schematic representation of our data preprocessing process. ECGs are segmented using a non-overlapping 30 s sliding window technique.

**Figure 3 sensors-23-08573-f003:**
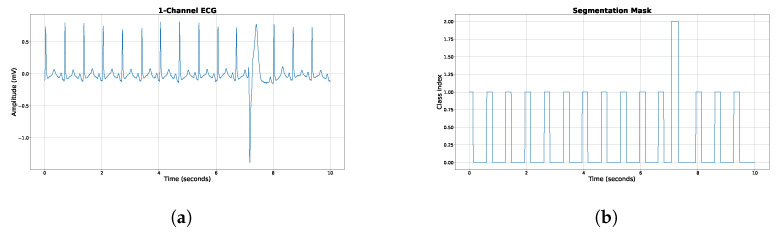
Example ECG from the MIT DB and the corresponding label mask. (**a**) Example ECG from the MIT arrhythmia dataset. A single PVC is located at around 7 s. (**b**) The corresponding mask containing background, normal beats, and PVC encoded with 0, 1, and 2, respectively.

**Figure 4 sensors-23-08573-f004:**
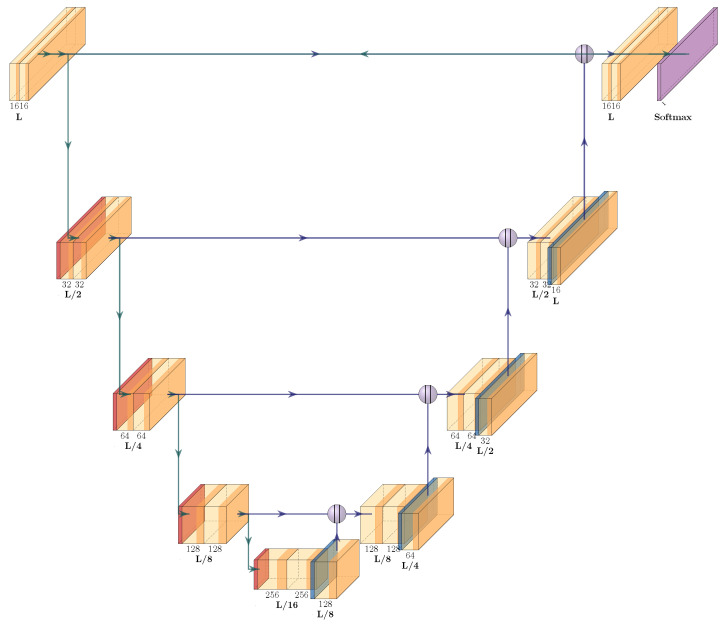
Our 1D U-Net architecture for ECG beat detection. Each block consists of a double convolution layer (1D Conv—Batch Norm—ReLU). For the decoder, we use simple linear upsampling layer instead of a transposed convolution layer. The output of our model is an L × C matrix, where L represents the length of the input ECG and C represents the three classes (no beat, beat, and PVC label).

**Figure 5 sensors-23-08573-f005:**
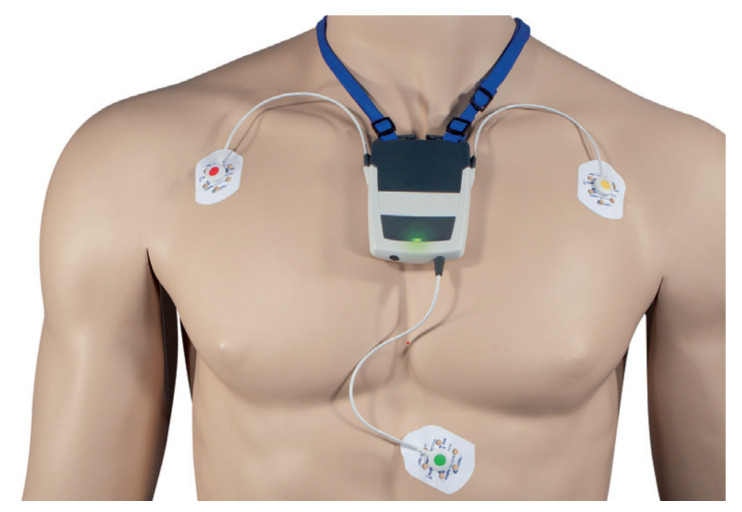
Custo med flash 500/510 3-Channel Holter used for data acquisition of our custom dataset.

**Figure 6 sensors-23-08573-f006:**
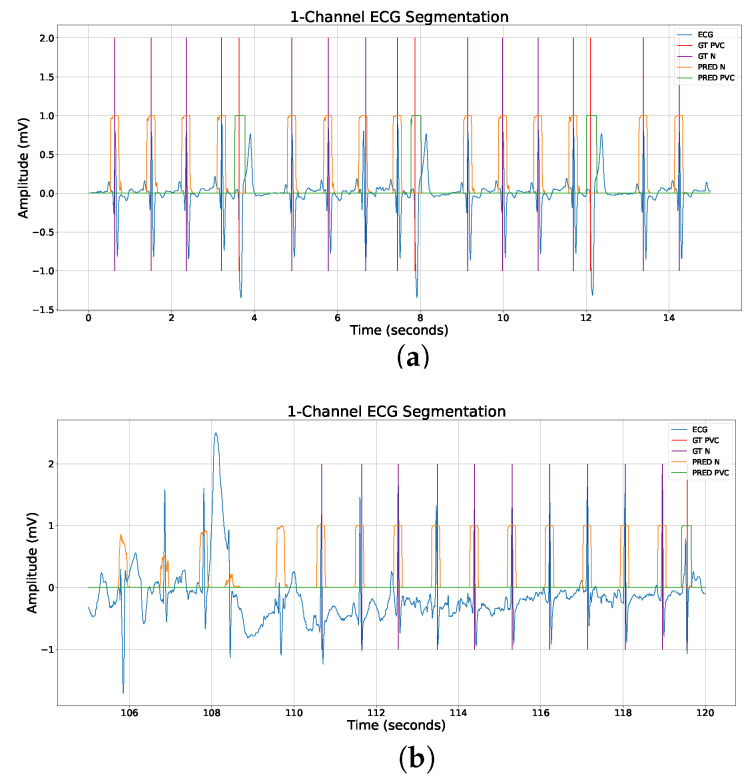
Clean vs. noisy ECG segmentation. The output masks show different behavior for clean and noisy ECG samples and require different post-processing treatment. (**a**) Example ECG from our custom dataset segmented into background, normal, and PVC beats. Three PVC beats are detected. (**b**) Example noisy ECG from our custom dataset segmented into background, normal, and PVC beats. A single PVC is located around 119 seconds. The normal beat class mask (orange) is noisy at the beginning due to uncertainty.

**Figure 7 sensors-23-08573-f007:**
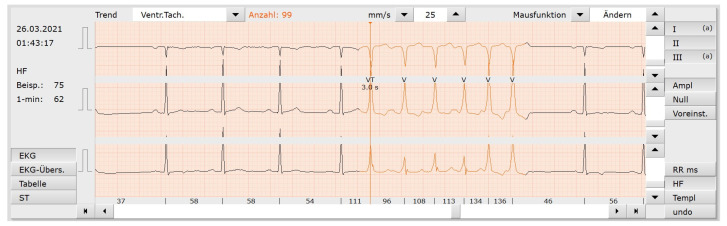
Screenshot from the custo diagnostic software. The model is enabled to predict normal beats (black) and ventricular tachycardia (beats labeled as V—orange).

**Figure 8 sensors-23-08573-f008:**
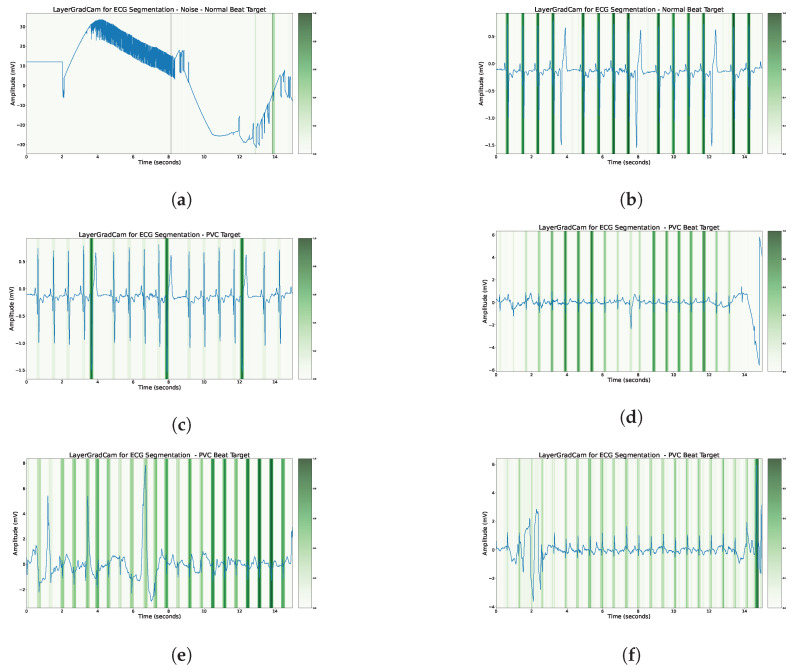
Examples from our custom medical test dataset utilizing LayerGradCAM attributions from the concluding decoder convolutional layer. A more pronounced attribution in a specific class is represented by a deeper shade of green. The R peak frequently emerges as the most salient feature. (**a**) The model’s sensitivity to patterns that mirror conventional QRS regions. (**d**) Surrounding beats influence the model’s decision regarding the PVC beat. (**e**,**f**) The model’s methodology of assessing the full segment prior to determining the PVC beat. Notably, these decisions are incorrect. (**a**) Activations for the normal beat class under noisy conditions (false positive). (**b**) Activations for the normal beat class. (**c**) Activations for the PVC beat class. (**d**) The PVC beat is not directly considered for the final decision. (**e**) Activations for the PVC beat class (false positive around 7 seconds). (**f**) Activations for the PVC beat class (false positive at the end of the segment).

**Table 1 sensors-23-08573-t001:** Comparison of normal beat detection algorithms on the MIT dataset with and without VFib (adapted from [13]).

Algorithm	Dataset	Se (%)	PPV (%)	F1
Pan and Tompkins [14]	MIT-BIH	99.76	99.56	99.66
Christov [15]	MIT-BIH	99.74	99.65	99.69
Chiarugi et al. [16]	MIT-BIH	99.76	99.81	99.78
Chouakri et al. [17]	MIT-BIH	98.68	97.24	97.95
Elgendi [18]	MIT-BIH	**99.78**	**99.87**	**99.82**
BeatLogic [13]	MIT-BIH	99.60	99.78	99.69
Liu et al. [19]	MIT-BIH	99.00	99.20	99.10
He et al. [20]	MIT-BIH	99.56	99.72	99.64
Martinez et al. [21]	MIT-BIH VFib excluded	**99.80**	**99.86**	**99.83**
Arzeno et al. [22]	MIT-BIH VFib excluded	99.68	99.63	99.65
Zidelmal et al. [23]	MIT-BIH VFib excluded	99.64	99.82	99.73
BeatLogic [13]	MIT-BIH VFib excluded	99.60	99.90	99.75

**Table 2 sensors-23-08573-t002:** Comparison of PVC beat detection algorithms on a subset of 11 MIT records (adapted from [13]).

Algorithm	Dataset	Se (%)	PPV (%)	F1
de Chazal et al. [24]	MIT-BIH 11	77.5	90.6	83.5
Jiang and Kong [25]	MIT-BIH 11	94.3	95.8	95.0
Ince et al. [26]	MIT-BIH 11	90.3	92.2	91.2
Kiranyaz et al. [2]	MIT-BIH 11	95.9	96.2	96.0
Zhang et al. [27]	MIT-BIH 11	97.6	97.6	97.6
BeatLogic [13]	MIT-BIH 11	97.9	98.9	98.4
Liu et al. [19]	MIT-BIH (22 records)	91.6	95.6	93.6

**Table 3 sensors-23-08573-t003:** Count and description of ECG beat types in the custo med training dataset.

Symbol	Beat Description	Count
N	Normal	3,361,174
V	Premature ventricular contraction	163,592

**Table 4 sensors-23-08573-t004:** Count and description of ECG beat types in the Icentia 11k dataset.

Symbol	Beat Description	Count
N	Normal	2,061,141,216
S	Premature or ectopic Supraventricular beat	19,346,728
V	Premature ventricular contraction	17,203,041
Q	Undefined: unclassifiable beat	676,364,002

**Table 5 sensors-23-08573-t005:** Patient counts by diagnosis in the St. Petersburg INCART dataset.

Diagnosis	Patients
Acute MI	2
Transient ischemic attack (angina pectoris)	5
Prior MI	4
Coronary artery disease with hypertension	7
Sinus node dysfunction	1
Supraventricular ectopy	18
Atrial fibrillation or SVTA	3 (2 with paroxysmal AF)
WPW	2
AV block	1
Bundle branch block	3

**Table 6 sensors-23-08573-t006:** Count and description of ECG beat types in AHA DB.

Symbol	Beat Description	Count
N	Normal	174,260
V	Premature ventricular contraction	16,296

**Table 7 sensors-23-08573-t007:** Count and description of ECG beat types in the MIT Arrhythmia dataset.

Symbol	Beat Description	Count
N	Normal	100,718
V	Premature ventricular contraction	7009

**Table 8 sensors-23-08573-t008:** Count and description of ECG beat types in the custo med test dataset.

Symbol	Beat Description	Count
N	Normal	39,133
V	Premature ventricular contraction	4576

**Table 9 sensors-23-08573-t009:** Results of our model for normal and PVC beat detection.

DB	${\mathrm{QRS}}_{\mathrm{Sp}}$	${\mathrm{QRS}}_{\mathrm{Se}}$	${\mathrm{PVC}}_{\mathrm{Sp}}$	${\mathrm{PVC}}_{\mathrm{Se}}$	${\mathrm{QRS}}_{\mathrm{BA}}$	${\mathrm{PVC}}_{\mathrm{BA}}$
MIT DB	0.997	0.999	0.926	0.966	0.998	0.946
MIT 11 DB	0.999	0.999	0.976	0.991	0.999	0.986
AHA DB	0.992	0.997	0.972	0.857	0.995	0.915
NST DB	0.954	0.893	0.936	0.881	0.924	0.909
CST DB	0.983	0.999	0.950	0.973	0.991	0.962
CSTStrips DB	0.993	0.997	0.960	0.932	0.995	0.946

Sp denotes specificity, Se denotes sensitivity, and BA denotes balanced accuracy.

## Data Availability

The data presented in this study are available on request from the corresponding author. The data are not publicly available due to privacy reasons.

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
