# Peer review of "End-to-End Premature Ventricular Contraction Detection Using Deep Neural Networks"

_sensors, 2023, doi:10.3390/s23208573_

Round 1

Reviewer 1 Report

Minor editing of English language required.

Author Response

Thank you very much for the review.
Attached you find our response. We updated the manuscript based on your suggestions.

Best regards

Dimitri Kraft

Reviewer 2 Report

The authors propose an algorithm for detecting premature ventricular contractions based on single-channel ECG recordings. The basic U-Net architecture was adopted for the underlying classification problem. In order to reduce the influence of easy negative samples, a composite loss function was introduced, which is a combination of the classical dice loss and the focal loss. Different ECG databases were used for training and testing the proposed method, excluding data leakage not only at the subject level but also at the level of the measurement setup. This simultaneously improves the robustness of the proposed method. The paper is well-written, well-structured, and the English phrasing is clear. However, a revision is necessary before the paper is published:

Comments:

  1. I appreciate the progressive literature review in Section 2, which not only provides an overview but also draws conclusions about current challenges, future research goals, and the limitations of state-of-the-art methods, along with their origins. I suggest adding another aspect of black box AI methods: the lack of interpretability. This is a crucial criterion that hinders the transition of AI-based solutions into clinical use cases. There have been attempts to open the black box of AI algorithms. One such attempt is applied by the authors in Section 5, where the network’s final decision is analyzed based on the input gradients (see e.g. [1]). Another approach is to define transparent network architectures/blocks/layers in which the activation and the trainable parameters themselves have a physical meaning. Ref. [2] is a good example, where the same PVC classification problem was considered. I recommend elaborating on this topic and including the suggested references.

  2. I really appreciate that, unlike many papers related to AI applications, this work describes the decisions made during the design of the architecture. For instance, the authors provide the reasoning behind using a 1D U-Net model, and the size of the receptive field that corresponds to the convolution blocks is well explained. However, I do not understand why simple linear up-sampling layers were used in the expanding path instead of the usual transposed convolution layer. Does it support the interpretability of the model’s output, or does it simply reduce the computational complexity?

  3. I truly appreciate the authors' effort in interpreting the trained U-Net model's output. However, this part of the study is not complete. In fact, only a few examples are provided in Section 5, but their consequences are not discussed. I strongly recommend extending the analysis by addressing the following questions:

    1. Which part of the ECG contributes the most to making the decision?

    2. What happens in the case of misclassified beats? Provide figures (similar to Fig. 6) for false negative examples and analyze the results.

    3. Provide LayerGradCam figures for noisy recordings as well. This would provide insight into why the model's performance is far from optimal in the case of noise.

Suggested references

[1] T. Bender et al., "Analysis of a Deep Learning Model for 12-Lead ECG Classification Reveals Learned Features Similar to Diagnostic Criteria," in IEEE Journal of Biomedical and Health Informatics, doi: 10.1109/JBHI.2023.3271858.

[2] Kovács, P., Bognár, G., Huber, C., Huemer, M., VPNET: Variable Projection Networks, International Journal of Neural Systems (IJNS), 2022, vol. 32, no. 1, pp. 2150054:1-19. doi: 10.1142/S0129065721500544

Author Response

(The authors gave the same response as above.)

Reviewer 3 Report

In this article, the authors show a method to detect normal heartbeats and Premature Ventricular Contractions (PVCs). I have some questions and comments:

1)      L24  “function independently of cloud resources.”L50:“. “A standout aspect of our approach is its operation solely on the existing hardware of physicians, thus rendering cloud support redundant” L:116” Moreover, our model operates independently of cloud resources, ensuring real-time detection with improved data privacy.”

I don't think anyone considers that your work should have been implemented in the cloud as a first option. It would rather be a second option. Since the amount of data is not high and its processing is not high, it does not make sense to use the cloud. So why consider in your comments as  advantageous  something that is not a first option and is not going to be used?

2)      Figure 1: You show an example of normal heartbeats and PVC. However, in this example, the detection of PVC is very simple since it is enough to apply a threshold. Can you also show another figure in which PVC is difficult to detect using a threshold?

3)      Figure 2: Could you improve the text size?

4)      L43: Robustness to Noise: But if you have multiple leads, can't it be an advantage to filter noise effectively?

5)      L83: “The team employed 10% of the PVC beats from the MIT dataset for training purposes, the very dataset they subse-quently used for validation.” I understand that they used the same data set for learning and for validation, is that it? And what happened to the 90%?

6)      L 107: “. Training and testing datasets are not separated.” In general, you can use the same amount of data between learning and validation when the amount of data is small through a cross validation process. This is correct if, despite the low amount of data, they are representative. With a sufficient amount of data, learning and validation is carried out. The test is only necessary if you have a new set of data acquired under conditions different from those of learning. For example, learning is done using signals with one type of noise and subsequently a test is done with signals with a different kind of noise. Noise can be filtered out in both situations and thus the test is likely to have the same result as learning. However, if filtering is not applied, the noise will be different and the test prediction can be bad. Therefore, there will be differences between learning and test. So if you have found articles where  “Training and testing datasets are not separated.” , possibly using good quality signals (e.g. no noise) with cross-validation could be sufficient to demonstrate good generalization. What do you think?

7)      On the other hand, in relation to noise, it does not seem that you have opted for any filtering technique to obtain noise-free ECG signals and thus achieve good quality signals. Do you have any explanation?

8)      L199: You are using N as input length. However, you are also using N in different tables.

Do you use a sliding window in your algorithm? What is the input length used?

9)      L262: In table 3, there is an example of the amount of data available. 163,592 PVC and 3361174 normal … There are other databases. A priori, 163,592 PVC + PVC from other databases is a fairly high number. And to create a balanced set it would be enough to select an equivalent number of Normal data. However, you have done "data augmentation". I'm not saying it's not positive, but was this augmentation necessary? Have you evaluated whether this augmentation improves the results? On the other hand, could you quantify how much PVC and Normal data have been used for learning and testing from all the databases + augmentation?

10)   Section 5: “Model Interpretability”: I think this title should be model explainability. Interpretability is when the result is consistent using different explanations. What do you think?

11)   In the discussion, you can make a comparison between your results and those shown in section 2.1 and 2.2..

12)   To improve, do you think that working with several leads would be positive to correct the noise significantly? In this way, signals would always be of good quality  and the model would make better predictions

Author Response

(The authors gave the same response as above.)

Round 2

Reviewer 2 Report

The Authors answered all of my questions, and revised the paper according to the reviewers' comments.

Reviewer 3 Report

Thank you for your responses